# Effects of Thermal and High-Pressure Processing on Quality Features and the Volatile Profiles of Cloudy Juices Obtained from Golden Delicious, Pinova, and Red Delicious Apple Cultivars

**DOI:** 10.3390/foods10123046

**Published:** 2021-12-08

**Authors:** Claudia Maria Liberatore, Martina Cirlini, Tommaso Ganino, Massimiliano Rinaldi, Silvia Tomaselli, Benedetta Chiancone

**Affiliations:** Department of Food and Drug, University of Parma, Parco Area delle Scienze, 27/A, 43124 Parma, Italy; clm.liberatore@gmail.com (C.M.L.); tommaso.ganino@unipr.it (T.G.); massimiliano.rinaldi@unipr.it (M.R.); silvia.tomaselli10@gmail.com (S.T.); benedetta.chiancone@unipr.it (B.C.)

**Keywords:** apple juices, high-pressure processing, physicochemical characteristics, quality parameters, volatile profile

## Abstract

In this study, juices extracted from three apple cultivars (Golden Delicious, Pinova, and Red Delicious) were stabilized by means of thermal treatment (TT) and high-pressure processing (HPP, 600 MPa 3 min); pH, total titratable acidity, total soluble solids content, color, and viscosity, as well as volatile profile, were investigated. Qualitative characteristics (pH, titratable acidity, colorimetric parameters, viscosity, and volatile profile) results were significantly influenced by both cultivars and treatments; for example, juice viscosity greatly increased after HPP treatment for Golden Delicious, and after both TT and HPP for Pinova, while no influence of stabilization treatment was registered for Red Delicious juices. Regarding the volatile profile, for Golden Delicious cultivar, HPP treatment determined an increase in volatile compounds for most of the classes considered, leading to a supposed quality implementation. For the other two cultivars, the stabilization treatment that better preserved the volatile profile was the HPP one, even if the results were quite similar to the thermal treatment. Further studies are needed to evaluate different time/pressure combinations that could give better results, depending on the specific apple cultivar.

## 1. Introduction

The World Health Organization (WHO, Geneva, Switzerland) recommends the consumption of at least 400 g/d of fresh fruits and vegetables [1]. In recent years, the consumers’ request is increasingly focused on minimally processed foods that maintain, as much as possible, the unchanged characteristics of the fruits and vegetables [2,3,4]. This consumer request clashes with the negative effect on juice quality of the traditional conservation methods, such as thermal treatments; indeed, fruit juices are pasteurized to prolong their shelf life, killing harmful microorganisms, and preventing cloud loss, inactivating heat-stable pectinmethylesterase (PME) [5]. Pasteurization, despite being very efficient, reduces the juice’s freshness perception [6] and changes its color [4]. Therefore, the juice industry, supported by technological investigation, is in constant research of conservation methods to obtain juices in which the organoleptical and nutritional values are very close to freshly squeezed juices [4,7]. Among innovative non-thermal treatments, high-pressure processing (HPP) is taking hold; this technology, though not involving the use of high temperatures, inactivates bacterial cells, yeasts, and molds, minimally affecting the organoleptic aspects of quality, such as texture, color, and flavor [8]; for these reasons, generally, HPP-treated foods are considered superior to thermal-treated ones, in terms of sensorial and nutritional quality [9,10,11,12].

Among fruit juices, apple juice is one of the most requested by consumers and, for this reason, is produced in Europe, United States, and Japan [12,13]. Nowadays, as in the past, but now with a scientific basis, it is well known that apples consumed fresh or transformed have a high impact on the human health, with a very strong antioxidant activity, decreasing lipid oxidation, and lowering cholesterol. These properties are linked with bioactive compounds, as flavonoids and phenolic acids, and with fiber, comprising pectins [13]. Several studies are focused on everything goes around the apple fruit and the numerous processes applied to transform it, without losing its important organoleptic and nutritional characteristics [12]. Relevant research has reported how thermal pasteurization and HPP technology differentially affect the main quality characteristics of apple juice [14,15,16,17]. Moreover, even if the aromatic profile of apple juice is influenced by both thermal and HPP treatments, the HPP technology guarantees a better preservation of the volatile compound composition, maintaining it in a manner comparable to that of the untreated matrix [12]. Indeed, due to the high temperatures involved in the process and the subsequent Maillard reaction, thermal pasteurization can lead to the formation of new volatile compounds and, therefore, to a changing in the final juice flavor and color, as reported in juices obtained from Pink Lady, Granny Smith, and Jonagold apple varieties [12]. The better quality of HPP-treated apple juices is not incontrovertibly demonstrated; indeed, it has been reported that the response of the apple juice to preservation treatment is a genotype-dependent response, so much so that, in some cases, such as in the New Zealand Jazz apple variety, thermal and HPP-treated juices showed comparable aromatic profile and with results different to those non treated [9].

The lack of research on the interaction between stabilization treatment and apple genotype lead to carry on this study aimed to deepen knowledge on the influence of the starting material (three apple cultivars, known on the market for their versatile attitudes, Golden Delicious (GD), Pinova (PIN), and Red Delicious (RD)) and of two pasteurization treatments (thermal treatment (TT) and HPP), on some important physical and chemical quality features of apple juice.

## 2. Materials and Methods

### 2.1. Plant Materials and Juice Production

Three apple cultivars—GD, PIN, and RD—were selected for juice extraction and for further stabilization treatments. Apples, purchased at the GDO, were firstly weighed (g) (9 random picked fruits per each variety) in order to determine the juice extraction yield. GD apples presented lower weight levels (166.5 ± 13.4 g) than PIN and RD (292.7 ±17.2 g and 285.1 ± 8.1 g, respectively). The texture of apples was analyzed using a TA.XT2 Texture Analyzer equipped with a 245.2 N load cell (Stable Micro Systems, Godalming, UK), a force resolution equal to 0.01 N, and an accuracy value of 0.025%. The parameters were quantified using the application software provided (Texture Expert for Windows, version 1.22). The puncture tests were performed on the equatorial part of the fruit (10 fruits were analyzed for each variety) using a 3 mm diameter cylinder probe at a speed of 1 mm s^−1^. The maximum penetration force (Hardens, given in N) from the force vs. the time curves was obtained from the penetration test. The non-destructive elastic deformation of the sample was also obtained (Slope, given in N mm^−1^) from the force vs. the distance curves. Hardness of apples resulted significant different among all varieties: GD (11.7 ± 0.9 N), RD (13.8 ± 1.8 N), and PIN (16.9 ± 2.0 N).

Apples of each variety were separately freshly squeezed to extract the juice by means of a domestic juice extractor (Centrikal metal, Ariete, Firenze, Italy); the juice extraction yield was, as follows: 70.8 ± 2.7%, 59.2 ± 3.4%, and 53.8 ± 3.2% for GD, PIN, and RD, respectively. In order to avoid juice oxidation, 500 mg/L of ascorbic acid were added to the juice just after the extraction. Juices were then packed in plastic bottles and immediately thermal (12 min at 87 °C) and HPP (3 min at 600 MPa) treated. Samples of all juices were stored for physicochemical analysis at −20 °C.

### 2.2. Physicochemical and Chemical Juice Characterization

pH was measured in triplicate by means of a pH meter (Jenway, Staffordshire, UK). The level of total soluble solids (TSS) was measured as Brix degrees by a refractometer (Optika, Ponteranica, Bergamo, Italy). Finally, total titratable acidity (TTA), expressed as malic acid concentration (g/L), was determined by titration with NaOH 0.1 N of 10 mL of juice properly diluted in water, adding phenolphthalein as indicator [18].

The rheological parameters of apple juices were determined in triplicate with the rotational rheometer ARES-TA^®^ (Advanced Rheometric Expansion System, TA Instruments, New Castle, DE, USA) and the Orchestrator TM software was used. The study was performed at 25 °C and at shear rates between 10 and 300 s^−1^ with the geometry of a concentric cylinder (Couette). The sensors had the following dimensions: diameter of the cup = 34 mm, concentric cylinder diameter = 32 mm, and length = 33 mm.

Colorimetric parameters of apple juice samples were evaluated with image analysis, as follows: samples were scanned by means of a desktop flatbed scanner (Hewlett Packard Scanjet 8200, Palo Alto, CA, USA) at 236 pixels per cm (600 dpi of resolution; true color–24 bit), equipped with a cold cathode lamp for reflective scanning. All images were scanned at the same conditions; during image acquisition, the scanner was held in a black box, in order to exclude surrounding light and external reflections. Flatbed scanner color (R, G, and B) was characterized and corrected as previously reported by N’Dri et al. (2010) [19]. For all tests, a total of 5 juices were analyzed for each variety.

### 2.3. Fruits and Juices Volatile Fraction Evaluation

Volatile fraction analysis was carried out both on apples and juices. Measures of 1 g of apple pulp and 2 mL of juice were collected for the analyses, conducted in duplicate.

All the analyses were conducted using a Thermo Scientific Trace 1300 gas chromatograph coupled to a Thermo Scientific ISQ single quadrupole mass spectrometer, equipped with an electronic impact (EI) source (Thermo Fisher Scientific, Waltham, MA, USA), following the methods described by Ricci et al. (2018, 2019) [20,21]. Briefly, extraction of the sample head space was performed using a Divinylbenzene–Carboxen–Polydimethylsiloxane (DVB–Carboxen–PDMS)-coated fiber (50/30 μm; Supelco, Bellefonte, PA, USA). The samples were previously equilibrated for 15 min at 40°, then the fiber was inserted in the head space for 30 min. After that, the desorption was conducted at 250 °C for 2 min, in splitless mode. Analytes were separated on a SUPELCOWAX 10 capillary column (Supelco, Bellefonte, PA, USA; 30 m × 0.25 mm × 0.25 μm), applying a temperature gradient. Temperature started from 50 °C for 3 min, and it was increased by 5 °C/min up to 200 °C, maintaining these final conditions for 12 min with a total run time of 45 min.

All the signals were acquired in full scan mode (40–500 *m/z*) and identified both by the calculation of linear retention indices (LRIs) on the basis of the retention times of a solution of C8–C20 alkanes analyzed under the same gas-chromatographic conditions applied for the sample, and by their registered mass spectra and subsequent comparison with instrumental libraries (NIST 14). Moreover, the volatile compounds were semi-quantified thanks to the addition of a reference (Toluene; 100 μg/mL in 10 mL of aqueous solution).

### 2.4. Statistical Analysis

For the fruits, one-way ANOVA was carried out to evidence the differences among cultivars, per each parameter evaluated; Tukey’s test (*p* < 0.05) was used for mean separation (SYSTAT 13.1, Systat Software, Inc.; Pint Richmond, CA, USA).

For the juices, per each parameter evaluated, two-way ANOVA was used to evaluate the influence of two factors—“Cultivar” (CV) and “Treatment” (T)—and mean separation was performed with Tukey’s test (*p* ≤ 0.05) (SYSTAT 13.1, Systat Software, Inc., Pint Richmond, CA, USA).

In addition to the stress analogies and the differences among the juices, the data obtained from volatile profile characterization were statistically elaborated by SPSS Statistics 26.0 software (SPSS Inc., Chicago, IL, USA), performing a principal component analysis (PCA) carried out using (as variables) the detected concentrations of each volatile. PCA was performed using covariance matrix and two factors were extracted.

## 3. Results and Discussion

### 3.1. Physicochemical and Chemical Characterization of Juices

Physicochemical parameters of juices were determined, in terms of pH, titratable acidity, soluble solid content, colorimetric parameters, and viscosity.

Measuring pH—giving relevant information on the juice acidity—is important to characterize the product flavor and to evaluate the subsequent technological processes. Statistical analysis of juice pH, a significant interaction of the two factors—cultivar and treatment—was performed. Indeed, considering the starting condition, in NT juices, PIN showed the statistically lowest pH, as already reported [22]; after the stabilization treatments, RD and PIN juice pH remained constant, while GD juice pH decreased, becoming statistically comparable with PIN pH (Table 1).

Effectively, only for GD, statistically analysis evidenced significant differences among treatments, with thermal-treated juices with a pH value statistically lower than those NT, and the HPP ones with an intermediate behavior (Table 1). It has been already reported that, in juices and pulp obtained from other fruits, HPP treatment increased the ionic dissociation constant of water and weakened acids [23], which determine the increase of H^+^ ions [24]; however, for other fruit juices, the HPP treatment is not as incisive, as reported for litchi-based mixed fruit beverage [25] and from PIN and RED juices. The thermal treatment does not have an unambiguous effect on fruit juices either, as several reports have indicated that thermal pasteurization is irrelevant on pH juice [26,27,28]. Results reported in this study, regarding the effect of thermal treatment on RD juice, are confirmed by Yang et al. (2019) [28], in the same apple cultivar. The lowering effect of thermal and/or HPP treatment on apple juice pH value could be genotype dependent, as reported by Yi et al. (2017) [12].

Together with the pH, the titratable acidity value contributes to the definition of juice flavor; the statistical analysis evidenced how “Cultivar” was the factor that mainly influenced this parameter. Specifically, it seems that RD juice has a titratable acidity significantly lower than GD and PIN (2.8 g/L malic acid vs. 4.8 g/L malic acid for both) (Table 1). TTA values recorded in this study are in line with those reported in previous studies regarding the TTA in non-treated RD juices [28], as well as being in line with those that did not evidence any significant influence of stabilization treatments—neither thermal nor HPP [12,28].

Sweetness is another important aspect of apple juice flavor. The total soluble sugar content (expressed as °Brix) of the apple juices analyzed in this study are comparable to those reported in the literature [28]. Statistical analysis did not detect any significant differences for the factors considered, meaning that the soluble solid content of apple juices has not been affected either by the preservation treatment nor by the cultivar, and resulted in the range 11.8–12.7 °Bx (Table 1). The same behavior has been reported by previous studies, carried out on the influence of several preservation treatments on physicochemical properties of apple juices [27,29,30].

The colorimetric parameters were significantly influenced by cultivar, treatment, and interaction between them; as expected, GD presented the highest L* value with the brightest juice [31], with a significant increase after thermal treatment, as reported by Krapfenbauer et al. (2006) [32]. RD juices presented the highest positive a* and the lowest b* values with the most intense red color, as expected, with a significant reduction for the first and significant increase for the latter for HPP and TT (Table 1). As stated by Falguera et al. (2012), changes on color parameters caused by high-pressure processing of apple juice are different depending on cultivar [33].

Viscosity values (Figure 1) of apple juices presented different trends for the three studied cultivars: viscosity increased after both TT and HPP for GD juices, decreased after both TT and HPP for PIN juices; finally, RD juices did not show any significant change in values after stabilization treatments. The behavior observed for the three cultivars may be due to the different pulp composition; indeed, since changes in viscosity may be due to pectin degradation, carried on by the enzymes polygalacturonase (PG) and pectinmethylesterase (PME), in GD—a cultivar with high galacturonic acid and pectins in its water fraction [34]—HPP treatment determined an increase in viscosity, while the behavior of RD and PIN samples could be related to molecular weight, conformation, degree of esterification, and charge densities of pectins from the specific cultivars. An increased viscosity, after HPP treatment, has been reported also in carrot juice, in which this phenomenon was attributed to changes in particle size and agglomeration [35], and in tomato puree, in which the inactivation of PG enzyme was observed [36]. Therefore, also for the parameter viscosity, the individuation of the right combination preservation treatment/apple cultivar is a key factor.

### 3.2. HS-SPME/GC-MS Volatile Fraction Characterization of Fruits and Juices

A total of 64 different volatile compounds, pertained to different chemical classes as esters, aldehydes, alcohols, terpenes, ketones, and other compounds, such as hydrocarbons, were identified in the head space of the fruits and juices (Table 2). A number of 49 of the 64 different aromatic molecules were identified in the head space of the fruits, while 51 were found in the volatile fractions of the juices, with 36 common compounds.

Considering the amount of the different volatiles found in the fruit head space, no significant differences were observed among the three cultivars (Figure 2); however, some differences can be noted for single volatiles (Appendix A). The aromatic compounds have a different distribution in the three apple varieties, as highlighted in Table 3. GD fruits were characterized by the presence of heptanal, pentylhexanoate, and hexylcaprylate; only in the PIN apple volatile profile it was possible to detect β-myrcene and 2-hexenyl acetate; while prenylcaproate and phenylethyl alcohol were observed only in the RD fruit volatile fraction (Table 3).

The head space of the juices obtained from the three cultivars separately presented, to a significant degree, the same volatiles of the starting matrices. Among all the 51 detected volatiles, several of them (36) had primary flavor compounds, as they were already present in the aromatic fraction of the fruits, while other molecules mainly derived from the process (Table 3). Some volatiles present in the aromatic fraction of fresh fruits were lost during the pressing phase, as were some esters, such as isoamylcaproate and hexylcaproate. Other aromatic compounds are generated by the extraction process, as by the stabilization treatments, especially aldehydes, such as furfural, benzaldehyde, and 2,4-decadienal, and alcohols, such as butanol. This latter was present in the starting juices of all the three cultivars, but it was lost after treatments in PIN and RD. The loss of alcohols could be ascribed to heating and to high pressure, applied for the stabilization of the products [49]. In addition, some volatiles resulted characteristic of the single cultivars, as heptanal detected in GD fruit and juice aromatic fractions (Table 3). Statistical analysis showed that, in juices, there was a significant interaction between the considered factors—cultivar and treatment. Specifically, for the NT juices, the total volatile amount resulted almost three times higher in PIN and RD cultivars compared with GD (Figure 3); in the thermal-treated juices, RD showed a total volatile content two folds higher than GD. Finally, among the HPP-treated juices, GD products were those with the highest total volatile amount. Considering the different cultivars, the TT did not seem to affect the aromatic molecule content, especially in PIN and RD cultivars, while HPP treatment determined an increase in the total volatile content in GD (Figure 3). This trend may depend both on the starting fruit variety and on the initial juice characteristics. It was indeed observed that in juices rich in aromatic components, such as those obtained squeezing Pink Lady and Jonagold apples, thermal pasteurization lead to an increase in volatiles compared with HPP treatment [12]. In our case, the volatile content in PIN and RD was not affected by either TT or HPP, while in GD juice, both treatments augmented the volatile quantity.

Among the chemical classes detected, aldehydes were mainly represented by hexanal and 2-hexenal, with typical herbal and apple notes and characteristic of apple juice flavor. Statistical analysis did not evidence any difference among these volatile contents in fruits, while in juices, a significant interaction between factors was noted. In fresh juices (NT), aldehyde concentration resulted lower in GD juice (1.80 ± 0.0 ppm) than in PIN and RD juices (5.6 ± 1.2 and 6.0 ± 1.4 ppm, respectively); this trend could be due to a more pronounced oxidation of fatty acids in PIN and RD. No differences were observed among the cultivars within TT and HPP-treated juices. Moreover, in GD juices submitted to HPP processing, a significant increase in total aldehyde content (6.6 ± 0.5 ppm) was observed compared with NT and TT (1.80 ± 0.0 ppm and 1.7 ± 0.9 ppm, respectively). This phenomenon could be mainly ascribed to a rise in hexanal and 2-hexenal concentration. Similar observations were noted in a previous study on New Zealand Jazz apple juices submitted to different stabilization treatments, in which the amount of 2-hexenal was higher in HPP-processed juices compared with the heated products [9].

Other typical apple flavor compounds are esters with fruity aromatic notes; among those, hexyl n-valerate, hexyl caproate, butyl caproate, and hexyl butyrate were the more abundant in fruits (Appendix A). All these volatiles have been identified as characteristic of different apple cultivars’ flavor [50,51,52]. Regarding juices, the statistically lowest ester quantity was found in GD fresh and pasteurized juices in respect to the other cultivars (Appendix A). Among the stabilization treatments, the TT induced an increase in total ester amounts in all the considered cultivar volatile fractions: in PIN and RD juices, the ester content was statistically more affected by heating in respect to GD products, while HPP treatment maintained, statistically unchanged, the initial ester concentration (Figure 3). The behavior of these components seemed to be related to the cultivar, as observed in previous studies in which the content of hexyl acetate contributed to differentiate apple juices on the basis of the variety [49].

No significant differences were detected among the total alcohol amount calculated for the three cultivars. The main representative compounds were hexanol and 2-hexen-1-ol with herbal and leafy notes, as shown in a previous study [21]. HPP treatment induced a significant decrease in total alcohol content only in RD juices, compared with NT; among the treatments, TT and HPP processes leaded to a diminution of total alcohol amount, in both PIN and RD juices, while in GD cultivar, alcohols were preserved after processing. This trend was mainly determined by hexanol, while 2-hexen-1-ol content was diminished by processing (Appendix A). While Reid et al. (2004) hypothesized that a decrease in alcohols could be caused by heating [49], other studies showed a rise of these compounds in the volatile fraction of apple juice, after the thermal treating, in comparison with the fresh and the HPP-processed product [12].

Among terpenes, α-farnesene resulted the most representative compound, both in fruit and juices (Appendix A), as reported for several cultivars as “Mela rossa Val Metauro” [53], Red Delicious, Mc Inthosh, Gala, and Empire [54]. Results obtained for juices showed a statistically significant interaction between factors, cultivar, and treatment, for total terpene amounts and for α-farnesene. Among cultivars, only for GD a statistically higher concentration of terpenes was found in HPP-treated product (1.23 ± 0.6 ppm), while only with HPP did the GD samples show a statistically higher content of terpenes and α-farnesene compared with the other two cultivars (0.09 ± 0.0 ppm for PIN and 0.43 ± 0.2 ppm for RD) (Figure 3). The trend of α-farnesene is probably linked to fruit variety, as it was indeed demonstrated in previous studies; the amount of α-farnesene was fundamental in the classification of apple juice samples pertaining to different cultivars, such as Jonagold and Bramley [28]. Even though no statistically significant differences were found among fruit aromatic fractions, GD showed higher α-farnesene concentration than the other two cultivars (Appendix A). For this reason, HPP-treated GD juice presented a more elevated α-farnesene amount. However, this aromatic compound is lost during stabilization treatments, as verified by Reid et al. (2004) [49].

Finally, sulcatone resulted the only ketone maintained in juice aromatic fraction, with a statistically higher concentration in GD juice treated with high pressure (0.13 ± 0.0 ppm). Among HPP juices, the statistically lowest amount of sulcatone was observed in the RD cultivar (0.03 ± 0.0 ppm). While Yi et al. (2017) reported that the presence of sulcatone was not observed in HPP-treated juices pertaining to Pink Lady, Granny Smith, and Jonagold cultivars, in the present study, HPP treatment affected the concentration of sulcatone, depending on the cultivar considered [12].

To stress and confirm diversities and analogies among the differently treated juices and the three cultivars, obtained data were statistically elaborated using an unsupervised approach. Principal component analysis (PCA) was performed using the concentrations of common volatile compounds as independent variables. To this aim, two components were extracted and covariance matrix method was applied. The first two components accounted for 87% of the total variance, as shown in Figure 4. More in detail, PIN and RD juices were separated from the GD ones on the basis of component 2, confirming the peculiarities of each variety. GD samples were characterized by a higher content of aldehydes, such as heptanal and benzaldehydes, in comparison with PIN and RD. Among GD cultivar, while NT and TT juices present a similar volatile content, characterized by the scarce amount of 3-hexanol, HPP samples were differentiated from the firsts on the basis of component 1, and in particular by hexanal, 2-nonenal, isoamyl alcohol, and benzaldehyde concentrations that were higher after HPP processing. Concerning PIN and RD samples, NT juices were well separated from treated samples, on the basis of a more elevated number of aldehydes, mainly hexanal. In addition, TT and HPP samples were distinguishable from each other; this distinction was based on the different quantity of hexyl acetate, 2-ethyl-1-hexanol, isoamyl acetate, (E)-2-methyl-2-butenyl acetate, and isobutyl acetate, which were higher in TT juices in respect to the HPP ones. Moreover, while for GD, NT and TT samples were differentiated from HPP, for PIN and RD cultivars, NT and HPP samples were similar for the second component. In particular, they presented comparable amounts of octanal, hexyl-n-valerate, and 1-octen-3-ol. On the basis of statistical data elaborations, it is possible to conclude that thermal and high-pressure treatments have a different influence on the volatile profile of considered juices, and changes given by processing seemed to be cultivar-dependent, as also stated by Yi et al. in 2017 [12]. In our case, HPP preserved better the aromatic characteristics of PIN and RD cultivars in respect to GD, for which an increase in volatile profile was observed.

## 4. Conclusions

Stabilization treatments (thermal or high pressure) differentially influenced the quality features of apple juices (pH, titratable acidity, colorimetric parameters, viscosity, and volatile profile), depending on the starting genotype; indeed, the interaction between preservation treatment and starting material seems to be crucial in the quality definition of the final product. In this study, HPP treatment enriched in volatile components—such as hexanal—in Golden Delicious juice, and contributed to preserving the initial volatile juice characteristics of PIN and RD. All this considered, even if HPP treatment may be more expensive than thermal treatment, it is worth considering for the stabilization of apple juices, for the more suitable cultivar.

In conclusion, this study suggests that different stabilization treatments must be considered on the basis of the characteristics of the starting materials. More studies are needed to investigate the applications and modulations of new technologies, such as HPP, with the aim to maintain fruit peculiarities in derived juices.

## Figures and Tables

**Figure 1 foods-10-03046-f001:**
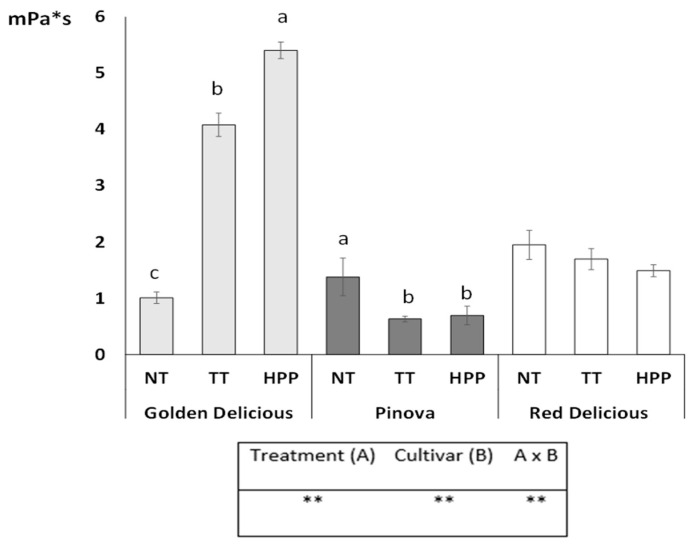
Viscosity values determined in apple juice samples. The means with different letters are significantly different (*p* < 0.05); stars indicate variable influence and strong interaction among considered factors.

**Figure 2 foods-10-03046-f002:**
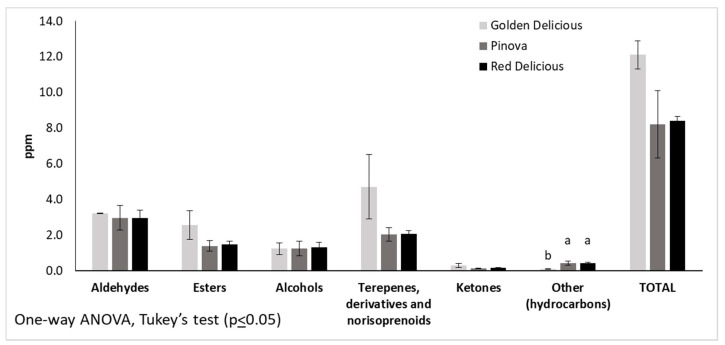
Volatile composition and amount of fruits pertaining to three apple cultivars. The means with different letters are significantly different (*p* < 0.05).

**Figure 3 foods-10-03046-f003:**
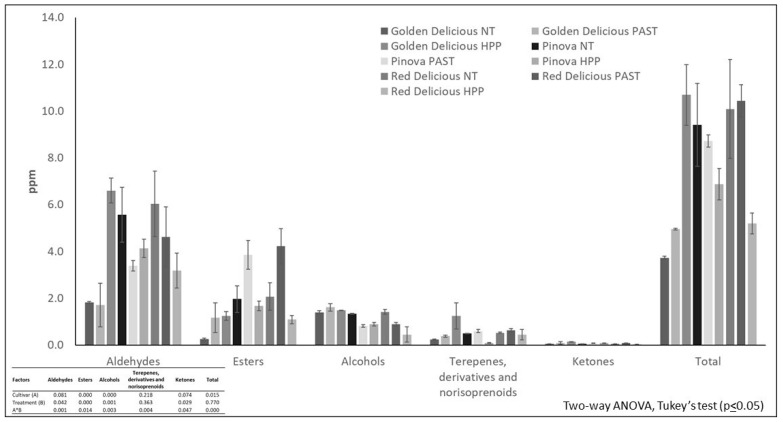
Volatile composition and amount of juices pertaining to three apple cultivars and different processes.

**Figure 4 foods-10-03046-f004:**
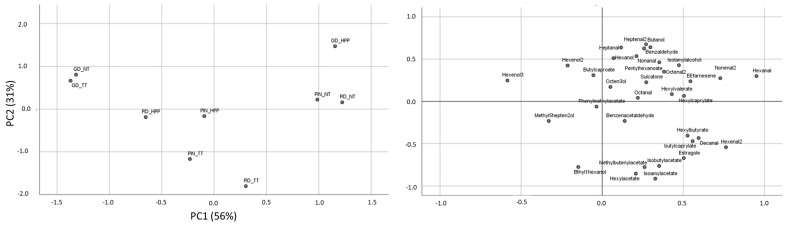
Scatter plot of scores from PC1 vs. PC2, obtained using the concentrations (ppm) of the commonly detected volatile compounds, observed for juice samples, and the relative loadings of the variables used.

**Table 1 foods-10-03046-t001:** Influence of cultivar and stabilization treatment on physicochemical parameters of apple juices.

Cultivar	Stabilization Treatment	pH ± SD	Total Titratable Acidity	Total Soluble Solids	Colorimetric Parameters		
			g/L Malic Acid ± SD	°Bx ± SD	L* ± SD	A* ± SD	B* ± SD
Golden Delicious	NT	3.7 ± 0.1	4.6 ± 0.7	12.2 ± 0.4	49.9 ± 0.7	0.1 ± 0.3	33.7 ± 1.5
	TT	3.2 ± 0.0	4.5 ± 0.2	12.1 ± 0.2	59.4 ± 0.4	−7.0 ± 0.0	43.0 ± 0.9
	HPP	3.4 ± 0.0	5.3 ± 0.2	11.8 ± 0.9	51.2 ± 0.2	−2.9 ± 0.2	39.1 ± 0.7
Pinova	NT	3.2 ± 0.1	4.4 ± 1.2	12.1 ± 0.2	47.2 ± 0.6	2.8 ± 0.2	36.7 ± 1.2
	TT	3.3 ± 0.0	4.8 ± 0.8	12.7 ± 0.4	44.6 ± 0.6	2.2 ± 0.4	34.3 ± 1.1
	HPP	3.3 ± 0.0	5.2 ± 0.4	12.5 ± 0.1	41.1 ± 0.5	3.4 ± 0.6	36.9 ± 1.1
Red Delicious	NT	3.7 ± 0.2	2.7 ± 0.4	12.6 ± 0.4	38.9 ± 1.2	11.1 ± 1.2	26.8 ± 0.9
	TT	3.7 ± 0.0	2.7 ± 0.1	12.2 ± 0.5	40.9 ± 0.8	4.4 ± 0.4	34.9 ± 0.8
	HPP	3.7 ± 0.0	3.0 ± 0.0	12.3 ± 0.2	37.6 ± 1.0	6.3 ± 0.9	32.0 ± 2.6
Cultivar (CV)		0.000	0.000	0.091	0.000	0.000	0.000
Stabilization Treatment (ST)		0.017	0.070	0.841	0.011	0.003	0.000
CV * ST		0.001	0.891	0.322	0.042	0.166	0.001

Two way ANOVA, Tukey’s test (*p* < 0.05). NT—non-treated juices; TT—thermal-treated juices; HPP—high-pressure processed juices; L*—lightness, A*—Red/Green Value, B*—Blue/Yellow Value. °Bx is the symbol used for Brix degree.

**Table 2 foods-10-03046-t002:** Volatile compounds of apple fruit and juice with the corresponding aromatic notes, calculated and tabulated LRIs, and references.

Peak Number	Identification	Matrix (A/J)	Aromatic Note	LRI Calc.	LRI Litt.	Reference
	Aldehydes					
1	Hexanal	A, J	Herbal	1078	1078	[21]
2	Heptanal	A, J	Herbal	1185	1187	[21]
3	2-Hexenal	A, J	Apple, green	1219	1220	[21]
4	Octanal	A, J	Aldehydic	1286	1294	[37]
5	2-Heptenal	A, J	Green	1320	1336	[38]
6	Nonanal	A, J	Aldehydic	1390	1390	[21]
7	2-Octenal	A, J	Green	1426	1438	[37]
8	Furfural	J	Bready, caramel	1467	1475	[39]
9	Decanal	A, J	Orange peel	1494	1492	[21]
10	Benzaldehyde	A, J	Fruity, almond	1524	1524	[21]
11	2-Nonenal	A, J	Green	1534	1546	[37]
12	Benzeneacetaldehyde	A, J	Green, honey	1653	1630	[40]
13	2,4-Decadienal	J	Orange, sweet	1812	1758	[41]
14	2,5-Dimethylbenzaldehyde	J		1873		
	Esters					
15	Isobutyl acetate	A, J	Sweet, fruity	1012	1005	[42]
16	Butyl acetate	J	Ethereal	1077	1105	[42]
17	Isoamyl acetate	A, J	Fruity, banana	1118	1113	[21]
18	Amyl acetate	J	Ethereal	1173	1176	[43]
19	Prenyl acetate	J	Sweet, fresh, banana	1244	1248	[43]
20	(E)-2-Methyl-2-butenyl acetate	A, J		1248	1250	[43]
21	Amylbutyrate	A	Sweet, fruity	1264		
22	Hexyl acetate	A, J	Fruit, herb	1270	1270	[21]
23	3-Hexenyl acetate	J	Green, fruity, apple	1313	1313	[44]
24	2-Hexenyl acetate	A	Green, fruity	1332	1329	[20]
25	Butylcaproate	A, J	Fruity, pineapple, apple	1408	1407	[21]
26	Hexylbutyrate	A, J	Green	1410	1411	[21]
27	Hexyl n-valerate	A, J	Fruity	1419		
28	Isoamylcaproate	A, J	Fruity	1453	1454	[21]
29	2-Hexenyl butyrate	A, J	Green, fruity, apricot	1471	1460	[43]
30	cis-3-Hexenyl 2-methylbutyrate	A	Fresh, green, apple	1475	1472	[43]
31	Pentylhexanoate	A, J	Sweet, fruity	1506	1505	[43]
32	Isobutyloctanoate	A	Fruity, green	1548	1550	[43]
33	Prenylcaproate	A	Cheesy	1577	1572	[43]
34	Hexylcaproate	A	Green	1603	1606	[21]
35	Butylcaprylate	A, J	Buttery	1607	1613	[43]
36	Ethyldecanoate	J	Sweet, waxy	1631	1645	[39]
37	2-Methylbutyl octanoate	A		1668	1657	[43]
38	(E)-2-Hexenyl hexanoate	A	Green, cognac	1665	1660	[43]
39	Phenylmethyl acetate	A, J	Sweet, floral	1763	1754	[44]
40	Hexylcaprylate	A, J	Green	1801	1803	[21]
	Alcohols					
41	Butanol	A, J	Fruity, wine	1140	1141	[21]
42	Isolamylalcohol	A, J	Alcoholic, whiskey	1205	1221	[21]
43	Prenol/2-Heptanol	J	Fruity/fresh	1317	1316	[43]
44	Hexanol	A, J	Herbal	1348	1349	[21]
45	3-Hexen-1-ol	A, J	Green, leafy	1381	1407	[42]
46	2-Hexen-1-ol	A, J	Leaf, green	1401	1402	[21]
47	1 Octen-3-ol	A, J	Earthy	1446	1455	[37]
48	1-Heptanol	J	Musty, leafy	1449	1460	[39]
49	6-Methyl-5-hepten-2-ol	A, J	Green	1459	1464	[45]
50	2-Ethyl-1-hexanol	A, J	Citrus	1484	1483	[21]
51	Octanol	J	Waxy	1553	1553	[21]
52	2-Octen-1-ol	J	Green, vegetable	1611	1611	[43]
53	Nonanol	J	Fresh, fatty, floral	1652	1657	[20]
54	PhenylethylAlcohol	A, J	Floral	1904	1931	[39]
	Terepenes, derivatives and norisoprenoids					
55	β-Myrcene	A	Spicy	1160	1168	[46]
56	Linalool	J	Floral	1542	1549	[39]
57	Caryophyllene	J	Sweet, woody	1592	1598	[47]
58	Estragole	A, J	Sweet, anise	1718	1685	[48]
59	(Z, E)-α-Farnesene	A	Sweet	1742	1737	[43]
60	(E, E)-α-Farnesene	A, J	Sweet, wood	1801	1764	[43]
	Ketones					
61	Sulcatone	A, J	Citrus	1335	1335	[21]
62	Butyrolactone	A	Bready	1631	1651	[37]
	Others					
63	Dodecane	A	Alkane	1197	1200	[43]
64	Tridecane	A		1292	1300	[43]

**Table 3 foods-10-03046-t003:** Presence/absence of the different aromatic compounds in the apple cultivars’ samples (fruits and juices).

Compounds	Golden Delcious				Pinova				Red Delicious.			
	Fruit	NT Juice	TT Juice	HPP Juice	Fruit	NT Juice	TT Juice	HPP Juice	Fruit	NT Juice	TT Juice	HPP Juice
Heptanal												
Octanal												
2-Heptenal												
2-Octenal												
Nonanal												
Benzaldehyde												
2-Nonenal												
Benzenacetaldehyde												
Isobutyl acetate												
(E)-2-Methyl-2-butenyl acetate												
Amylbutyrate												
2-Hexenyl acetate												
Hexylbutyrate												
Isoamylcaproate												
2-hexenyl butyrate												
cis-3-Hexenyl 2-methylbutyrate												
Pentylhexanoate												
Isobutyloctanoate												
Prenylcaproate												
Hexylcaproate												
butylcaprylate												
2-methylbutyl octanoate												
(E)-2-Hexenyl hexanoate												
Phenylmethyl acetate												
Hexylcaprylate												
Butanol												
Isolamylalcohol												
3-Hexen-1-ol												
2-Hexen-1-ol												
1 Octen-3-ol												
6-Methyl-5-hepten-2-ol												
2-Ethyl-1-hexanol												
PhenylethylAlcohol												
β-Myrcene												
Butyrolactone												

NT—non-treated juices; TT—thermal-treated juices; HPP—high-pressure treated juices. Small grids of different colors represent the existence of the substances.

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
