# Peer review of "Effects of Thermal and High-Pressure Processing on Quality Features and the Volatile Profiles of Cloudy Juices Obtained from Golden Delicious, Pinova, and Red Delicious Apple Cultivars"

_foods, 2021, doi:10.3390/foods10123046_

Round 1

Reviewer 1 Report

Introduction

  1. The effect of thermal and high pressure treatments on the volatile profile of apples could be introduced based on previous researches because your study mainly focused on this part and other basic physicochemical parameters only take up a small part of your manuscript.

  1. Line 48-52, the sentence needs to be revised.

Materials and methods

  1. The HS-SPME/GC-MS method of analysis needs to be described clearly in the manuscript, though you cited literature.

Results and discussion

  1. Table 1, one-way ANOVA may be also needed. Significant differences and insignificant differences should be presented between NT, TT, and HPP within the same cultivar. Also, significant differences and insignificant differences should be presented between different cultivars within the same treatment. I suggest letters ‘a, b, c…’ and ‘A, B, C…’ can be used for them.

  1. Line 157-160, usually, high galacturonic acid content represents high pectin content in food. High pectin content normally induced higher viscosity of food. Besides, Xie et al. reported higher viscosity of pectin from potato peel after high pressure treatment and it is not the viscosity of potato peel as you described. In my opinion, the viscosity changes of juice after thermal or high pressure treatment mainly depend on pectin concentration in water-soluble phase and structure changes after treatment (high pressure may active PME enzyme and subsequently induced a de-esterification of pectin. Low ester pectin can cross-link with Ca2+ which induces a network increasing the viscosity of the system. The thermal treatment may induce a β-elimination of pectin. High pressure and thermal may induce more pectin transfer from insoluble to soluble fraction.). There are several possible factors affecting the viscosity of juice after treatment. So this section may need to be discussed in an appropriate way.

  1. For volatile fraction, if possible, I suggest a Multivariate analysis such as PCA could be presented, because there are a huge number of data concerning volatile changes between different treatments and cultivars and it is difficult to compare them in a table. Multivariate analysis can find out key components distinguishing different samples.

  1. Line 295-296, from the result and discussion of this study, I cannot find the better preservation and enhancement of the physicochemical properties of apple juice after HPP treatment. This sentence needs to be revised.

Author Response

Introduction

  1. The effect of thermal and high pressure treatments on the volatile profile of apples could be introduced based on previous researches because your study mainly focused on this part and other basic physicochemical parameters only take up a small part of your manuscript.

We tried to implement this section (lines 58-70) even if, at present, few studies are focused on the evaluation of volatile profile of differently processed apple juice.

  1. Line 48-52, the sentence needs to be revised.

We tried to clarify these sentences (lines 47-57).

Materials and methods

  1. The HS-SPME/GC-MS method of analysis needs to be described clearly in the manuscript, though you cited literature.

We added all the information concerning HS-SPME/GC-MS method applied for the analyses in paragraph 2.3.

Results and discussion

  1. Table 1, one-way ANOVA may be also needed. Significant differences and insignificant differences should be presented between NT, TT, and HPP within the same cultivar. Also, significant differences and insignificant differences should be presented between different cultivars within the same treatment. I suggest letters ‘a, b, c…’ and ‘A, B, C…’ can be used for them.

Two-way ANOVA considers the effect of two categorical factors, and the effect of the categorical factors on each other, analysing and presenting the data resorting to one-way ANOVA, and so analysing every single factor independently, in our opinion, will reduce the information obtained, because the influence of the factor not considered will be missed.

  1. Line 157-160, usually, high galacturonic acid content represents high pectin content in food. High pectin content normally induced higher viscosity of food. Besides, Xie et al. reported higher viscosity of pectin from potato peel after high pressure treatment and it is not the viscosity of potato peel as you described. In my opinion, the viscosity changes of juice after thermal or high pressure treatment mainly depend on pectin concentration in water-soluble phase and structure changes after treatment (high pressure may active PME enzyme and subsequently induced a de-esterification of pectin. Low ester pectin can cross-link with Ca2+ which induces a network increasing the viscosity of the system. The thermal treatment may induce a β-elimination of pectin. High pressure and thermal may induce more pectin transfer from insoluble to soluble fraction.). There are several possible factors affecting the viscosity of juice after treatment. So this section may need to be discussed in an appropriate way.

We tried to reformulate the discussion about all the physico-chemical parameters, without taking into consideration conductivity, and in particular viscosity changes (Paragraph 3.1).

  1. For volatile fraction, if possible, I suggest a Multivariate analysis such as PCA could be presented, because there are a huge number of data concerning volatile changes between different treatments and cultivars and it is difficult to compare them in a table. Multivariate analysis can find out key components distinguishing different samples

We performed a PCA on data obtained for juices, in order to confirm or better underline the differences and analogies found on the basis of other statistical tests. (4040 – 431, Figure 4).

  1. Line 295-296, from the result and discussion of this study, I cannot find the better preservation and enhancement of the physicochemical properties of apple juice after HPP treatment. This sentence needs to be revised.

We tried to revise the sentence (lines 425-436).

Reviewer 2 Report

The study aimed to study the effects of different processing technology on physical-chemical properties and volatile profile of apple juice from three apple cultivars. The research contents in this manuscript was normal, and lack of innovation. The maturity is important to the quality of apple juice, which is varied greatly when the maturity change. In this manuscript, there are no any measures to take to guarantee the maturity. In my opinion, it did not meet the requirement of Foods.

  1. The introduction part is poor written, please rewrite.
  2. Why you determine the texture of apples and electrical conductivity of apple juice? I think they are not important for evaluate the juice quality.
  3. What meaning of the different color in table 3?

Author Response

The study aimed to study the effects of different processing technology on physical-chemical properties and volatile profile of apple juice from three apple cultivars. The research contents in this manuscript was normal, and lack of innovation. The maturity is important to the quality of apple juice, which is varied greatly when the maturity change. In this manuscript, there are no any measures to take to guarantee the maturity. In my opinion, it did not meet the requirement of Foods.

Apples were recovered from the market, at the ripening stage and comparable to each other. The degree of maturity has not been determined. The focus of the work was not the evaluation of the characteristics of the males, but of the juice and above all of the effect of the different treatments on the aromatic composition.

  1. The introduction part is poor written, please rewrite.

We improved the introduction part.

  1. Why you determine the texture of apples and electrical conductivity of apple juice? I think they are not important for evaluate the juice quality.

We agree with the reviewer. We discussed more properly the part regarding physical and chemical parameters (Paragraph 3.1), and since conductivity did not describe important information about juices, we decided not to consider it.

  1. What meaning of the different color in table 3?

Table 3 reported the presence/absence of volatile compounds. The colors are used to differentiate the samples from each other

Reviewer 3 Report

The important results of the variation of volatile compounds in apple juice treated with temperature, and those treated by HPP should be included in the introduction.

In the results, it is suggested to review significant differences, and the ppm referred to toluene generates confusion, it must be expressed more clearly

In the analysis of volatile compounds, 1 g of pulp and 2 mL of apple juice were used, the variation of the headspace volume must be considered to buy the results. In this case it is better to express the results divided by dry weight. Table 3 could be replaced by the response area / g dried apple, of each volatile in each treatment, figure 3, must be modified.

A discussion should be included about the influence of volatile compounds that show significant variations (eg threshold of perception) or about the compounds that are effectively important in the perception of apple juice.

Author Response

The important results of the variation of volatile compounds in apple juice treated with temperature, and those treated by HPP should be included in the introduction.

To better introduce the changes in volatile profile given by processing, we added some information based on literature (lines 63-73, Introduction).

In the results, it is suggested to review significant differences, and the ppm referred to toluene generates confusion, it must be expressed more clearly

In order to underline possible differences and analogies among juices, based on the three different cultivars and the different preservation treatment, we performed a PCA analysis and we added the obtained results (4040 – 431, Figure 4).

Volatile compounds were semi-quantified on the basis of an internal standard; after different experiments with different standard volatiles, Toluene was chosen because it presented a retention time different from those of all the other detected molecules. Volatile concentrations are expressed in ppm (mg/L) and they have been calculated on the basis of Toluene, but they are not referred directly to it. The applied method of semi-quantification is indeed the so called “Internal standard method”.

In the analysis of volatile compounds, 1 g of pulp and 2 mL of apple juice were used, the variation of the headspace volume must be considered to buy the results. In this case it is better to express the results divided by dry weight. Table 3 could be replaced by the response area / g dried apple, of each volatile in each treatment, figure 3, must be modified.

Table 3 reported the presence/absence of volatile compounds, not their amounts. The table was built with the aim to better describe which varietal aromatic compounds could be found also in the final juice, and at the same time which of them were lost, or formed during processing.

In addition, in the calculation of the final concentration of each detected molecule, the initial amount of sample was taken into account.

A discussion should be included about the influence of volatile compounds that show significant variations (eg threshold of perception) or about the compounds that are effectively important in the perception of apple juice.

In some cases it was indicated whether the compounds found were actually typical of the aromatic fraction of the apple (i.e. 2-hexenal).

It is difficult to indicate if an aromatic compound can actually be perceived or not, because this does not depend only on the threshold value (not tabulated for all volatile compounds) but on the aroma value, which is calculated on the basis of threshold value. Since, threshold values are not reported for all the volatiles compound found in the present work, it was not possible to do this consideration.

Reviewer 4 Report

The manuscript entitled "Effects of thermal and high-pressure processing on quality features and volatile profile of cloudy juices obtained from Golden Delicious, Pinova and Red delicious apple cultivars" contains new and significant information in the field. However, there were some issues that need to be addressed before publication. It is important to state clearly implications for research, practice and society. In my opinion, it is relevant to emphasize the importance of molecules from fruits in maintaining balanced health. It has been mentioned information, but I recommend to extend. In this regard, I kindly recommend the next paper to be consulted for the introduction section: DOI: http://dx.doi.org/10.5772/intechopen.91218.

For the section "volatile extraction" I suggest including supplementary information regarding the technique used (parameters, temperatures, strokes, time, costs, etc.) in the identification/characterization of volatile compounds from samples.

Results and Discussions could be improved by studying other papers in the field.

Is it also important to mention that apple varieties have different chemical characteristics which influence the sensory profile of the finished product.

A brief comment by the Authors on the strengths and weaknesses of the study presented. It did not provide any significant management recommendations for farmers or industry. The HPP treatment is reliable from an economical point of view?

Please refine the language carefully and avoid repetition of some words (e.g. to better, etc.).

Author Response

The manuscript entitled "Effects of thermal and high-pressure processing on quality features and volatile profile of cloudy juices obtained from Golden Delicious, Pinova and Red delicious apple cultivars" contains new and significant information in the field. However, there were some issues that need to be addressed before publication. It is important to state clearly implications for research, practice and society. In my opinion, it is relevant to emphasize the importance of molecules from fruits in maintaining balanced health. It has been mentioned information, but I recommend to extend. In this regard, I kindly recommend the next paper to be consulted for the introduction section: DOI: http://dx.doi.org/10.5772/intechopen.91218

Even if our work is more focused on the changes that different preservation treatments, thermal and HPP processing, can cause on volatile profile of apple juices, with a particular emphasis on different cultivars, we agree with the reviewer. Apple is indeed a fruit with several nutritional benefits given by different bioactive compounds, as other fruits rich in polyphenols (i.e. berries, etc.). For this reason, we tried to stress more this concept, adding some information in the introduction section (lines 48-53)

For the section "volatile extraction" I suggest including supplementary information regarding the technique used (parameters, temperatures, strokes, time, costs, etc.) in the identification/characterization of volatile compounds from samples.

We added all the information concerning HS-SPME/GC-MS method applied for the analyses in paragraph 2.3.

Results and Discussions could be improved by studying other papers in the field.

We tried to improve the “Result and Discussion” session, adding comments on evaluation of physico-chemical parameters measured on juices (without taking into account conductivity), also comparing our results with other studies. To improve more the discussion about volatile profile determination, we also added results about an additional statistical elaboration of data (PCA). In addition, few studies are focused to study the changes in volatile fraction caused by HPP treatment in apple juice, so to the best of our knowledge we cited all the papers that described these phenomena.

Is it also important to mention that apple varieties have different chemical characteristics which influence the sensory profile of the finished product.

We agree with the reviewer. To better underline the differences both among the three considered cultivars as between the two different treatments, and to demonstrate that the aromatic profile of the final products is linked to the initial characteristics of the juice, that are cultivar dependent, we performed a PCA analysis, that helps to explain what described above. (lines 388-414)

A brief comment by the Authors on the strengths and weaknesses of the study presented. It did not provide any significant management recommendations for farmers or industry. The HPP treatment is reliable from an economical point of view?

We agree with the reviewer, HPP treatment may be more expensive than the thermal one, but it seems to better preserve the aromatic characteristics of the starting apple juice (for two of the considered cultivars) or to increase the volatile profile content (for GD), avoiding in part the formation of Maillard reaction products. We tried to add this consideration in Conclusion section.

Please refine the language carefully and avoid repetition of some words (e.g. to better, etc.).

Authors revised the language, improving the form of language

Round 2

Reviewer 1 Report

The manuscript has been improved, I suggest it can be accepted after language modification.

Reviewer 3 Report

Changes and modifications made improve the manuscript